# Mechanism of LH release after peripheral administration of kisspeptin in cattle

**Carlos E. P. Leonardi, Rodrigo A. Carrasco, Fernanda C. F. Dias, Eric M. Zwiefelhofer, Gregg P. Adams, Jaswant Singh⊙ \***

Department of Veterinary Biomedical Sciences, Western College of Veterinary Medicine, University of Saskatchewan, Saskatoon, Canada

\* jaswant.singh@usask.ca

**Data Availability Statement:** All relevant data are within the paper and its Supporting Information file.

## Abstract

Kisspeptin modulates GnRH secretion in mammals and peripheral administration of 10-amino acid fragment of kisspeptin (Kp10) induces LH release and ovulation in cattle. Experiments were done to determine if iv administration of kisspeptin will activate GnRH neurons (i.e., after crossing the blood-brain barrier) and if pre-treatment with a GnRH receptor blocker will alter kisspeptin-induced LH release (from gonadotrophs) and ovulation. In Experiment 1, cows (n = 3 per group) were given human-Kisspeptin10 (hKp10; 3 x 15 mg iv at 60-min intervals) or normal saline and euthanized 150 min after treatment was initiated. Every 20th free-floating section (50 µm thickness) from the preoptic area to hypothalamus was double immunostained to colocalize GnRH- (DAB) and activated neurons (cFOS; Nickel-DAB). Kisspeptin induced plasma LH release from 15 to 150 min (P = 0.01) but the proportion of activated GnRH neurons did not differ between groups (5.8% and 3.5%, respectively; P = 0.11). Immunogold electron microscopy detected close contacts between kisspeptin fibers and GnRH terminals in the median eminence. In Experiment 2, pubertal heifers (n = 5 per group) were treated with 1) hKp10 iv, 2) Cetrorelix (GnRH antagonist; im) + hKp10 iv or 3) saline on Day 6 of the follicular wave under low-progesterone condition. A rise in plasma LH concentration was detected from 15 to 240 min in the hKp10 group but not in cetrorelix or control group (P<0.001). Ovulations were detected only in the hKp10 group (4/5; P = 0.02). Cetrorelix treatment was associated with regression of the preovulatory dominant follicle and emergence of a new follicular wave 3.4±0.75 days after the treatment in all five heifers. Results support the hypothesis that the effect of peripheral kisspeptin is mediated downstream of GnRH synthesis and does not involve GnRH-independent LH release from gonadotrophs. Peripheral kisspeptin may release pre-synthesized GnRH from the nerve terminals in areas outside the blood-brain barrier.

## Introduction

Kisspeptin is a neuropeptide product of kiss-1 gene that is cleaved and/or degraded in 54-, 14-, 13- and 10-amino acid peptides [1]. Kisspeptin is highly expressed in the brain and mutations of kisspeptin receptor (Kiss1R), also named G-protein receptor-54 (GPR-54),

**Funding:** Research for this paper was funded by the Natural Sciences and Engineering Council of Canada Discovery grant (RGPIN-2017-05750) The funders had no role in study design, data collection and analysis, decision to publish, or preparation of the manuscript.

**Competing interests:** The authors have declared that no competing interests exist.

induce hypothalamic hypogonadism and impaired reproductive maturation in humans and mice [2,3]. The C-terminal end of amino acid kisspeptin sequence is essential for complete activation of Kiss1R, and the shortest sequence form (10-amino acids) has higher bio-potency than longer forms [1,4]. Hypothalamic GnRH neurons express Kiss1R in rats [5,6]. Furthermore, GnRH and kisspeptin immunoreactive cells are located in the preoptic area and in the hypothalamus in mice, primates, sheep [7–9] and cattle [10], and its projection are detected in close association in the median eminence. Peripheral injections of kisspeptin induce luteinizing hormone (LH) secretion and ovulation in several mammalian species [11–15] including cows [16,17] and seasonally anestrus sheep [11]. Further, intravenous administration of kisspeptin-10 induces the release of GnRH into the hypophyseal portal circulation in sheep [6]. It is unknown if the 10-amino acid kisspeptin fragment (kisspeptin-10) is able to cross the blood-brain barrier to stimulate GnRH neurons after peripheral injection in cattle.

The DNA-binding oncogene protein, c-FOS, changes gene transcription in response to cellular membrane signals and has been used as a biomarker to identify activated neurons after neuronal stimulation [5,18–20]. The majority of neurons do not express c-FOS under baseline conditions [19,21], however, the expression of the c-FOS gene increase dramatically after neuronal stimulation resulting in c-FOS protein production within 2–5 h, with a peak between 60–90 min [21,22]. Thus, dual immunohistochemistry to co-localize a protein target and c-FOS expression is an effective tool to determine the activation of specific neuronal populations. There was a high correlation between increased c-FOS expression in GnRH neurons and the LH surge in male and female rats and ewes [5,19,23], but not in monkeys [24]. Bypassing the blood-brain barrier by intra-cerebroventricular administration of kisspeptin-10 induced concurrent LH secretion and cFOS expression in 85% of rat GnRH neurons after 2 hours of administration [5] with some evidence of neuronal activation after subcutaneous administration [15]. However, results of a study in ewes suggest that intravenous kisspeptin-10 administration may not cross the blood-brain barrier because the peptide was not detected in the cerebrospinal fluid [12]. It is not yet known if peripheral kisspeptin can activate the GnRH neurons in cattle.

Endogenous and exogenous GnRH induces release and synthesis of LH from the pituitary gland in cattle [25,26] and administration of a GnRH antagonist prevented the LH surge [27]. Peripheral administration of kisspeptin increased plasma LH concentration in ovariectomized cows [28] and pubertal heifers [29], and induced ovulation under a low-progesterone milieu [16]. In a previous study, we detected a progressive increase in LH secretion within 15 minutes after repeated intravenous administration of kisspeptin-10 in cows with plasma progesterone below 1.7 ng/mL [16]. However, the mechanism by which peripheral administration of kisspeptin-10 induces LH release in cows is unclear. *In vitro* studies in horses and cattle raise the possibility of direct LH release from anterior pituitary cells by kisspeptin [30,31]. Whether kisspeptin-10 crosses the blood-brain barrier to stimulate GnRH neuronal cell bodies, acts on GnRH nerve terminals in median eminence, or the observed LH release is due to a direct effect on the pituitary gonadotrophs *in vivo* remains unknown.

The objective of the present study was to elucidate if peripheral administration of kisspeptin will increase LH secretion through activation of GnRH neurons, and to determine if pre-treatment with a GnRH receptor blocker will alter the pattern of kisspeptin-induced LH release and ovulation. We tested the hypotheses that: 1) administration of human kisspeptin-10, given as multiple intravenous doses to cows in proestrus, will activate GnRH neurons to produce cFOS; 2) pre-treatment with a GnRH antagonist (Cetrorelix) before kisspeptin treatment will suppress LH release and prevent ovulation in cattle.

## Material and methods

### Animals

Two experiments were conducted on non-pregnant cattle in the Fall season. Experiment 1 was conducted on lactating Holstein cows (n = 6; 674±7 Kg body weight) during October. Cows were kept in an outdoor pen with access to hay and water *ad libitum*, and milked twice daily. Experiment 2 was performed in November on Hereford crossbred heifers (n = 15; 500±24 Kg body weight, 17–18 month age) that had a corpus luteum in one of the ovaries at the start of experiment. The heifers were maintained in outdoors pens at the University of Saskatchewan LFCE Goodale Research Farm (52˚ north and 106˚ west). Heifers were fed barley silage and had hay and water *ad libitum*. A mineral salt block was available to heifers throughout the study period. All procedures were performed in accordance with Canadian Council on Animal Care and were approved by the University of Saskatchewan Protocol Review Committee.

### Kisspeptin-10 peptide

The human Kisspeptin-10 (hKP-10) peptide (YNWNSFGLRF-NH2) was custom synthetized at >95% of purity (MW: 1318.44 g/mol) by GenScript USA Inc, Piscataway, NJ, USA. The sequence is based on the predicted C-terminal region (112-121-NH2) of human metastin (Gen Bank accession # AY117143) and has been previously used in cattle [16,28,29]. The peptide was previously tested for solubility and was dissolved in ultrapure water at 10 mg/mL.

### Experiment 1: c-FOS expression after Kisspeptin-10 treatment during proestrus period

The ovarian status of cows (n = 6) was determined by transrectal ultrasonography using a 7.5 MHz linear-array transducer (MyLab 5, Esoate, Maastricht, Netherlands). Cows in which a corpus luteum (CL) was detected were selected and given 500 µg cloprostenol im (PGF2α; Estrumate, Merck Animal Health, Kirkland, QC, Canada) twice 12 hours apart to cause luteolysis and ovulation. Daily ovarian ultrasonography was performed to monitor follicle development and ovulation (Day 0). Ovulation was defined as the disappearance of a large follicle from one examination to the next, followed by the development of a CL. A luteolytic dose of PGF2α was given on Day 5.5 and repeated 12 hours later to create a proestrus phase. An indwelling jugular catheter was fitted in place 36 hours after first PGF2α im injection as described previously [32], and cows were assigned randomly to two groups: kisspeptin group (n = 3, 3 doses of 15 mg hKp10 iv at 60-min intervals), or control group (n = 3, 3 iv injections of 1.5 mL normal saline solution at 60-min intervals). The dose and frequency of treatment was determined based on our previous studies in cattle [16,17]. Blood samples were collected at 15 min intervals from -30 to 150 min (0 min = time of first injection) in heparinized tubes (Vacutainer, BD, Franklin Lakes NJ, USA). After the last blood collection, cows were euthanized by a bolus intravenous dose of sodium pentobarbital (Euthanyl Forte, 540 mg/mL, 1mL/5kg body weight, Bimeda-MTC Animal Health Inc, Cambridge, Ontario, Canada).

**Tissue collection and preparation.** After euthanasia, cow heads were removed at the level of third cervical vertebra, a plastic cannula was inserted into the lumen of left and right common carotid artery and held with hemostatic forceps for the perfusion procedure. The brain was perfused in situ with 4L of cold normal saline solution with 10 IU of heparin sodium per mL followed by 2L of 4% paraformaldehyde in phosphate buffered saline (PBS; 0.1M, pH = 7.4) using a peristaltic pump (~250 mL/min flow rate). Half of the volume was perfused through each artery (i.e., 2L cold normal saline and 1 L of 4% paraformaldehyde in the right carotid artery and the same volume through left carotid artery). Approximately 15 minutes

after completion of the perfusion fixation procedure, the whole brain was removed by cutting the skull bones. The mid-brain (rostral portion of the preoptic area to the mammillary body) was dissected and placed in 4% paraformaldehyde for 48 hours at 4˚C in the equivalent of 10 tissue-volumes. The brain tissue was cryoprotected by immersion in increasing concentration of sucrose solution in 0.1M PBS, beginning with 10%, 20% and 30% (w/v) until it sank. Finally, the tissue block containing the preoptic area (POA) and hypothalamus (approximately 40 mm length in rostral-caudal direction x 36 mm width x 40 mm height in dorso-ventral direction) was frozen at -80˚C for ≥48 hour until sectioning. Serial brain sections (coronal sections) were obtained with a cryostat microtome at 50 μm thickness. Brain sections were immersed immediately in cryoprotectant solution containing 30% glucose and 30% glycerol, placed in 2mL tubes and stored at -20˚C until further processing. The cryoprotectant solution remained liquid at this temperature. In total 405 to 486 sections were obtained per tissue block.

**Dual immunohistochemistry of GnRH and cFOS.** Every 20[th] free-floating section from POA to middle of mammillary body was processed by double immunostaining using two sequential avidin-biotin-peroxidase reactions optimized for thick (50μm) sections: nickel- diaminobenzidine reaction (purple-black; nuclear localization) for c-FOS and diaminobenzidine reaction (brown color; cytoplasmic localization in perikarya and nerve fibers) for GnRH. Since both antigens are in different cytosolic compartments, it was expected to have minimal cross-reactivity. All solutions were made in 0.1M phosphate buffered saline pH 7.4 (PBS) unless otherwise specified and sections were washed in PBS 3 times x 5min on a rocking platform between all steps. The serial sections were warmed up to room temperature for 20 min before removal of the cryoprotection solution. The samples were placed individually in six wells plate dishes and identity of the sections were maintained throughout the staining procedure. The brain slices were washed three times (once 30 min, twice 5 min) in PBS before the antigen retrieval by placing them in the hot water bath at 90˚C for 18min in 3mL of citrate buffer (0.01M, pH 6.0) containing 0.3% (v/v) Tween20. Following antigen retrieval, samples were allowed to cool for 20 min at room temperature and washed. The sections were incubated in a blocking solution (1% bovine serum albumin, 0.3% (v/v) TritonX) for 4h at room temperature on a rocking platform. Next, sections were incubated (without washing) in a polyclonal rabbit anti-c-FOS antiserum solution (1:30,000 dilution in blocking buffer; Catalog # ABE457, EMD Millipore, Germany) and incubated for 72 hours at 4˚C. Following the primary antibody incubation, the sections were placed in 3% hydrogen peroxide solution for 30 min to quench endogenous peroxidase. Then, the samples were incubated for 1h in the biotin-conjugated secondary antibody (Goat anti-Rabbit IgG (H+L) Biotin; Thermo Fisher Scientific C#31822) at a 1:500 dilution in PBS with 0.3% TritonX. Finally, the slides were immersed in a solution containing peroxidase-conjugated Streptavidin (Streptavidin-HRP, Jackson Immuno Research Inc. West Grove, PA, USA) diluted 1:10,000. The peroxidase activity was visualized by incubating the sections in a solution containing 2.5% nickel sulfate, diaminobenzidine (DAB) substrate (DAB, 3,3'-Diaminobenzidin; Sigma-Aldrich, Inc.) and hydrogen peroxide (0.05% v/v). Upon color development, sections were then washed, immersed in polyclonal rabbit anti-GnRH antiserum for the sequential staining (1:40,000, LR-5, a generous gift from Dr. R. Benoit, McGill University, Montreal, Canada) and incubated for 72 hours at 4˚C. The secondary antibody (Goat anti-Rabbit Biotin; Thermo Fisher Scientific Catalog # 31822) was used at 1:500 dilution for an hour. Finally, the slides were incubated in 1:10,000 peroxidase-conjugated streptavidin and the peroxidase activity was detected by the DAB-substrate (resulting in brown reaction). The free-floating sections were mounted on large glass slides (75 X 50 mm and 0.96 to 1.06 mm thickness; Corning Incorporated, USA). The slides were air dried at room temperature and coverslipped (48 X 65 mm, Number 1; Thermo Scientific, USA) using a xylene-based mounting medium (Eukitt, Sigma-Aldrich, Oukville, ON, Canada, #03989).

The number of GnRH perikarya (without cFOS-staining) and cFOS co-localized GnRH positive perikarya were recorded in different regions/neuronal nuclei of the POA and hypothalamus at 10x and 40x objective magnification. Every 20th serial section (i.e., at 1mm intervals) from the cranial POA to the mammillary body was analyzed.

**DAB (GnRH) and immunogold (kisspeptin) staining for transmission electron microscopy.** Midbrain sagittal sections containing the median eminence (n = 2 cows from the kisspeptin group) were processed for immunoelectron microscopy to determine the association between kisspeptin and GnRH nerve terminals. Enblock immunostaining of cryostat sections (50 μm thickness) was performed for GnRH using DAB procedure as describe in previous section. The median eminence containing DAB staining was dissected from the floating sections under a light microscope and further processed for LR white resin embedding. Samples were rinsed in sodium cacodylate buffer, and dehydrated through graded ethanol series and placed in LH white resin overnight for infiltration under black light. Ultrathin sections (60 to 80 nm thickness) were cut and placed on nickel grids. Samples were incubated in polyclonal rabbit anti-kisspeptin antiserum (1:10 dilution; AC566, a generous gift from Dr. Franceschini, INRA, Physiologie de la Reproduction et des Comportements, Nouzilly, France; [33]) and the antigen-antibody complex was detected with an anti-rabbit antibody conjugated to 10 nm gold particles (Goat anti-Rabbit Gold, Sigma-Aldrich, St. Louis, MO, USA, #G7402). Sections were observed under a transmission electron microscope at 80 kv (Hitachi HT7700).

**Antibody controls.** The controls were performed with four sections (two from preoptic area, and two containing arcuate nucleus) per brain. The specificity of the GnRH and cFOS (S1 Fig) antibodies were tested by pre-adsorption with the GnRH peptide (ab 120184; Abcam, Cambridge, MA, USA) and cFOS protein (AB56280-1002 Abcam, Cambridge, MA, USA), respectivelly. The cFOS and GnRH immunoreactive cells were not detected by using pre-adsorption antibody to peptide or protein. Also, cFOS and GnRH antibodies were omitted during the procedure with no resultant immunoreaction.

## Experiment 2: Effect of GnRH antagonist (cetrorelix) on response to kisspeptin treatment

Heifers (n = 15) were selected from a larger group based on the detection of a CL by transrectal ultrasonography. Immediately after ultrasound examination, heifers were given PGF2α twice at an interval of 12-hours. To test our hypotheses, we used the low-progesterone bovine model described previously (Fig 1) [16]. Briefly, the ovaries were examined daily by ultrasonography to detect ovulation. Three days after ovulation, follicles ≥5mm in diameter in both ovaries were ablated by transvaginal ultrasound-guided follicle aspiration, inducing the emergence of new follicular wave 1.5 day after the procedure [34]. After follicular ablation, a progesterone

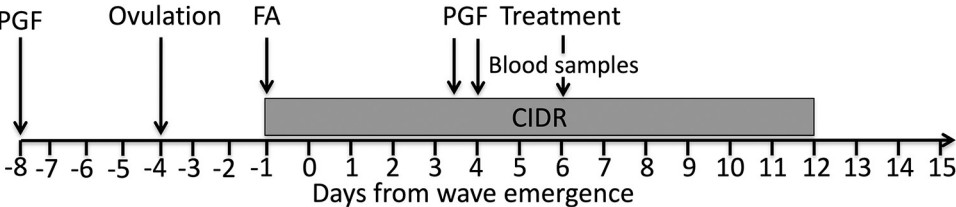

**Fig 1. Experimental model used in Experiment 2.** Emergence of a new follicular wave (Day 0) was induced by follicular ablation (FA) three days after ovulation, and an intravaginal progesterone-releasing device (CIDR) was inserted immediately after follicle ablation and left in place for 13 days. Luteolytic doses of a prostaglandin $F_{2\alpha}$ analog (cloprostenol) were given on Day 3.5 and 4. Ovarian ultrasonography was performed daily during the experiment. Blood collection and treatment administration was done on Day 6.

device was placed in the vagina (CIDR, Pfizer Canada, Inc., Montreal, Quebec, Canada). The day of wave emergence (Day 0) was defined as the day when the dominant follicle was first recorded (4 to 5 mm in diameter) with subsequent increase in diameter. Heifers were given PGF2α (500 µg cloprostenol im, Estrumate, Merck Animal Health, Kirkland, QC, Canada) on Days 3.5 and 4 (i.e., 8 and 8.5 days after ovulation) to cause luteolysis, however, the CIDR devices were left in place until Day 12 to maintain a low-progesterone milieu sufficient to prevent spontaneous ovulation. On Day 6, an indwelling jugular catheter was placed as described [32] and heifers were assigned randomly to three groups (n = 5 per group): 1) 3 iv doses of 15 mg human kisspeptin-10 at 1h intervals (Kp10 group), 2) pretreatment with single im dose of cetrorelix acetate (20 µg/Kg of body weight, GnRH antagonist—Sigma #C5249 diluted in 5% D-Manitol w/v, Sigma #M4125) followed 3h later with 3 iv doses of 15 mg human kisspeptin-10 at 1h intervals (Cetrorelix group), or 3) 3 iv doses of equivalent volume of normal saline at 1h intervals (Control group). Following treatment (on Day 6), ovarian ultrasound examinations were performed daily until the end of the experiment to detect ovulation of the extant dominant follicle (i.e., dominant follicle present at the time of treatment) or of the dominant follicle originating from a subsequent wave. The progesterone device was withdrawn on Day 12 (i.e., thirteen days after insertion) and ultrasound examinations were continued until ovulation was detected.

## Blood samples and hormone assays

Serial blood samples on the day of treatment were obtained at 15 min intervals from -30 to 150 min (0 min = time of first hKP-10 or saline injection) using an indwelling jugular catheter as described above in the experimental design section of previous experiment. All blood samples were collected in heparinized tubes (Vacutainer, BD, Franklin Lakes NJ, USA). Immediately after sampling, tubes were centrifuged at 1500 x g for 15 min, and plasma was separated and stored at -20°C.

The plasma samples from both experiments were analyzed for LH and progesterone concentration at the University of Wisconsin (Madison, WI, USA) in Dr O.J. Ginther's Research Laboratory. Plasma LH concentrations were measured with a validated radioimmunoassay for cattle [35] with modifications as reported [36]. Briefly, LH concentrations were measured in duplicate using USDA-bLH-B-6 for $^{125}$I-iodination and for preparing reference standards, and USDA-309-684P as the primary antibody (National Hormone and Pituitary Program, Torrance, CA, USA). The standard curve ranged from 0.078 to 20.0 ng/mL with sensitivity of 0.1 ng mL$^{-1}$. Intra- and inter-assay coefficients of variation and mean sensitivity were 6.23%, 12.24% and 0.03ng/mL, respectively. Progesterone concentrations were measured as described [37] in a single assay batch with a commercial solid-phase RIA kit containing antibody-coated tubes and $^{125}$I-labeled progesterone (ImmuChem Coated Tube progesterone 125 RIA kit, MP Biomedical, Costa Mesa, CA). The intra-assay coefficients of variation and sensitivity for progesterone were 11.97% and 0.06 ng/mL, respectively.

## Statistical analyses

Data analyses were performed using SAS (Statistical Analysis System, software package 9.4, SAS Institute Inc., Cary, NC, USA). In both experiments, single-point measurements (i.e., diameter of the dominant follicle at the time of treatment and 24 hours after treatment; diameter of the CL at the time of treatment and progesterone concentration at the time of treatment) were analyzed using one-way analysis of variance or Student's T-Test (for data with 2 treatment groups, Experiment 1). Statistical significance was assumed when the *P*-value was ≤0.05 whereas a tendency for a difference was assumed when the *P*-value was between >0.05 to

$\leq$0.1. Tukey's post-hoc test was used for multiple comparisons if the *P*-value for a test detected a difference. Ovulation rate was analyzed using the GLIMMIX procedure.

Analyses of repeated measures data (e.g., LH plasma levels, follicular dynamic) were performed using MIXED models procedure, in which treatment, time, and treatment-by-time interaction were tested and a repeat statement was included in the syntax (repeated days subject = cowID). Initial analyses tested five covariance structures (SIMPLE, CS, AR(1), ANTE (1), or UN) and the model with smallest AICC value was selected for final analysis. All values are reported as mean ± SEM.

## Results

### Experiment 1: c-FOS expression after Kisspeptin-10 treatment during proestrus

The diameter of dominant ovarian follicle and CL at the time of PGF2α injection (Day 6 of follicular wave) and the time of treatment (Day 7) did not differ between groups (Table 1). Plasma progesterone concentrations on Day 7 were also similar (P = 0.15) between the groups. Overall, the concentrations of plasma LH were higher after kisspeptin treatment (Fig 2A, P = 0.01) than the control group but individual variations were noted (Fig 2B).

The total number of GnRH immunoreactive perikarya (Fig 3A) did not differ between the treatment groups (kisspeptin: 44.33±7.3, control: 28.33±3.3; P = 0.11), nor did the proportion of GnRH perikarya that co-expressed c-FOS (Fig 3B) (kisspeptin: 5.8%, control: 3.5%; P = 0.11). Individual variations were recorded in the number of perikarya that express c-FOS only (Table 2).

Sagittal sections from tissues processed by immunoelectron microscopy demonstrated an close-appositions between kisspeptin and GnRH nerve terminals at the level of the median eminence. The secretory vesicles of GnRH were apparent as generalized electron dense areas (DAB reaction) within the terminal portion of GnRH axons while the Kp-immunoreactive nerve terminals showed aggregation of gold particles surrounding the GnRH nerve fibers (Fig 4).

### Experiment 2: Effect of GnRH antagonist on kisspeptin response

The diameter of the dominant ovarian follicle at the time of treatment (Day 6 of follicular wave) and 24h after treatment did not differ among the Kp10, Cetrorelix and Control groups (Table 3). Plasma progesterone concentrations were similar among groups at the time of treatment. The Kp10 group had higher plasma LH concentrations than the Cetrorelix and Control groups (P<0.001; Fig 5B). Plasma LH concentrations remained higher in the Kp10 group than

**Table 1. Ovarian and endocrine responses (mean±SEM) of lactating cows in proestrus treated with three iv injections of 15mg human Kisspeptin-10 or normal saline (control) at 1h intervals on Day 7 (Day 0 = day of wave emergence) in Experiment 1.** Data were compared by t-test.

| Endpoint | Control (n = 3) | Kisspeptin (n = 3) | *P-value* |
|---|---|---|---|
| Plasma progesterone concentration on Day 7 (ng/ml) | 1.74±0.11 | 0.96±0.43 | 0.15 |
| Dominant follicle diameter (mm): | | | |
| on Day 6 | 15.33±0.22 | 16.75±0.52 | 0.06 |
| on Day 7 | 16.08±0.58 | 18.08±0.91 | 0.14 |
| Corpus luteum diameter (mm): | | | |
| on Day 6 | 20.41±1.74 | 20.16±3.16 | 0.94 |
| on Day 7 | 16.91±0.08 | 16.51±1.52 | 0.79 |

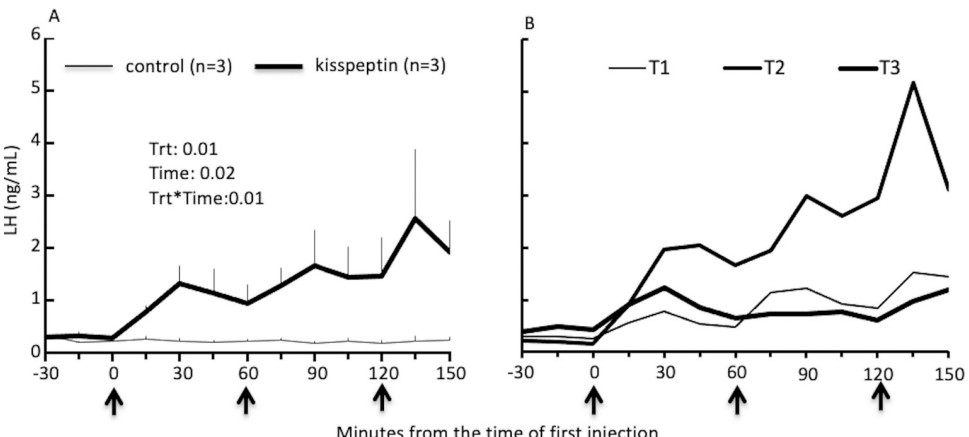

**Fig 2. Plasma LH concentration in the experiment 1.** A) Plasma LH plasma concentrations in lactating cows treated with three iv injections of 15mg human Kisspeptin-10 (kisspeptin; n = 3) or normal saline (control; n = 3) at 1h intervals on Day 7 (Day 0 = day of wave emergence) in Experiment 1. B) Plasma LH profiles of individual cows from the kisspeptin group illustrating similar pattern in two cows but varying magnitude of response among the three animals. Black arrows along the x-axes indicate the time of treatments.

the other two groups from 15 to 240 min (0 min = first injection of the treatment; Fig 5B). Ovulation of the extant dominant follicle was recorded in 4 of 5 heifers in the Kp10 group compared to 0 of 5 heifers in the Cetrorelix and Control groups (Table 3). A new follicular wave emerged immediately in all four heifers that ovulated after kisspeptin treatment (Fig 5A). Regression of the extant dominant follicle was recorded 3 to 4 days after treatment in the Cetrorelix group (Fig 5A). Emergence of a new follicular wave occurred 3.40±0.75 days after treatment in the Cetrorelix group (i.e., without ovulation of extant dominant follicle) compared to 2.75±0.75 days in the Kp10 group (after ovulation of extant dominant follicle in 4 of 5 heifers). In the Control group, the extant dominant follicles did not regress, and ovulated on Day 15 after CIDR removal (withdrawn on Day 12). All new dominant follicles from the Kp10 (n = 4) and Cetrorelix (n = 5) group ovulated on Day 15 after the CIDR removal (Day 12).

## Discussion

Our research group recently discovered that multiple intravenous doses of kisspeptin at short intervals induce LH secretion and ovulation of dominant follicle in cattle under low plasma progesterone environment [16]. Kisspeptin neurons in hypothalamus are known to modulate GnRH secretion in several species [11,14,15] but the mechanism of action of peripheral administration of kisspeptin on hypothalamic-pituitary gonadal axis is unclear in cattle. Results of this study document that kisspeptin given intravenously at 1h intervals induced LH secretion confirming our previous results, but that this effect was not mediated through activation of GnRH neurons (i.e., no change in proportion of GnRH cells expressing cFOS) in the preoptic area or hypothalamus of cows in proestrus (i.e., the most permissive stage for kisspeptin action). Further, pre-treatment with a GnRH antagonist abolished the LH releasing effect of kisspeptin and prevented ovulation, ruling out direct effect of kisspeptin on gonadotrophs. Close contact between kisspeptin fibers and GnRH-containing nerve terminals in the median eminence was detected, and collectively, results support the hypothesis that the effect of peripheral kisspeptin administration is mediated downstream of GnRH synthesis, perhaps by inducing release of pre-synthesized GnRH from the nerve terminals in areas outside the blood-brain barrier.

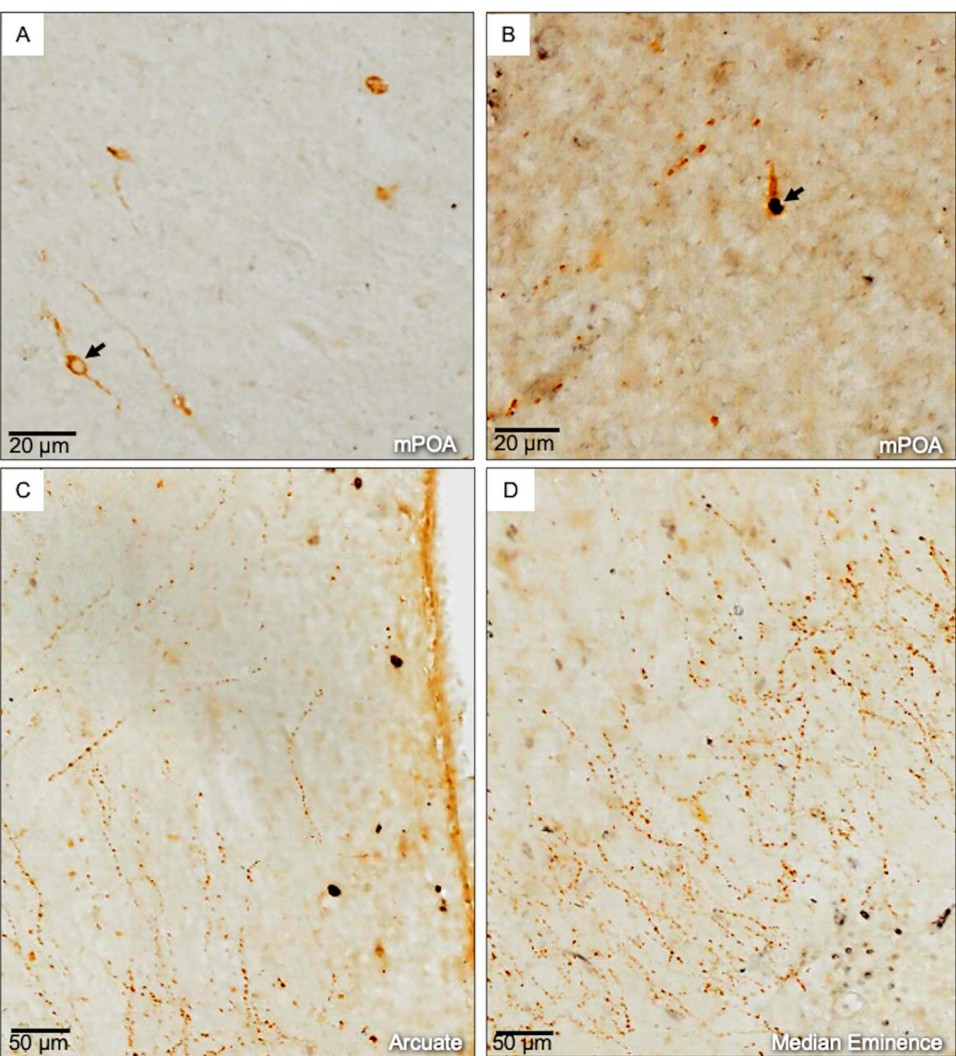

**Fig 3. Distribution of GnRH and c-FOS immunoreactive cells in the preoptic area and hypothalamus after administration of 3 doses of 15 mg human Kisspeptin-10 intravenously (at 1 hr intervals) during proestrous phase of estrous cycle in cows.** A) GnRH immunoreactive neurons (arrow) in the medial preoptic area (mPOA) nucleus showing cytoplasmic DAB (brown) staining. B) GnRH immunoreactive neurons that co-express c-FOS immunoreactive reaction (Nickel-DAB black color). Note that c-FOS is localized in the nucleus (arrow). C-D) GnRH (brown) and c-FOS (black) immunoreaction cells and nerve fibers in the arcuate nucleus and median eminence.

While repeated administration of human kisspeptin-10 induced LH release in proestrous cows, the treatment did not induce GnRH neuronal transcription. Our results are consistent with previous studies in that peripheral administration of kisspeptin increased plasma LH concentrations in rats, sheep, cattle, goats, and horses [11,14,15,38,39]. However, we were unable to detect activation of GnRH cell bodies (i.e., no differences in cFOS expression in GnRH neurons between groups) when hypothalamic tissues were examined 2.5 hours after kisspeptin treatment. The approach used in this study to evaluate neuronal activation has been applied to investigate a large number of drugs that were administrated by peripheral route and their effect in GnRH neurons including kisspeptin [23,40,41]. In addition, the advantage of using cFOS expression is that these proteins are located in the nucleus, identification of the phenotype of the activated neurons using substances located within the cytoplasm, such as GnRH, can be accomplished with standard double-labeling immunocytochemical techniques which is the

**Table 2.** Number of neurons expressing immunoreactivity for c-FOS positive perikarya (c-FOS), GnRH positive perikarya (GnRH) and GnRH neurons bodies that co-expressed cFOS protein (c-FOS and GnRH in the preoptic and hypothalamic area of cows 150 minutes after initiation of treatment with kisspeptin or saline (Time 0 = first administration).

| Cow | Group | Number of perikarya in preoptic area + hypothalamus | | |
|---|---|---|---|---|
| | | c-FOS | GnRH | GnRH+cFOS |
| C1 | control | 248 | 35 | 1 |
| C2 | control | 1860 | 25 | 0 |
| C3 | control | 3954 | 25 | 1 |
| T1 | kisspeptin | 599 | 58 | 4 |
| T2 | kisspeptin | 1546 | 42 | 3 |
| T3 | kisspeptin | 1031 | 33 | 1 |

case of current study [41,42]. Furthermore, the plasma LH concentrations was two to three folds higher at the moment of brain collection, and within of 150 minutes after kisspeptin treatment which is appropriate to detect cFOS expression [42]. The preovulatory LH surge in sheep was associated to cFOS expression in 30–40% of hypothalamic GnRH neurons [19]. In contrast, human kisspeptin-10 induced LH release after intracerebroventricular and intraperitoneal administration but only induced c-FOS expression after the central administration [43]. Surgical models in sheep have shown that kisspeptin was undetectable (by radioimmunoassay) in cerebrospinal fluid after intravenous administration [12] while intravenous administration of kisspeptin stimulated the release of GnRH into hypophysial portal blood circulation in sheep [12]. Currently, this approach is not technically feasible in cattle but a similar mechanism of action of kisspeptin is plausible. We interpret our cFOS findings to support the hypothesis that intravenously administrated kisspeptin does not cross the blood brain barrier in cattle similar to that in sheep [44]. In contrast to these results, subcutaneous administration of kisspeptin enhanced cFOS immunoreactity in 60% of GnRH neurons in male rat within 3 hours [15] indicating that species difference may exist in selective permeability or active transport of kisspeptin across the blood brain barrier.

One possible mechanism by which circulating kisspeptin may be able to elicit the observed LH release would be by GnRH-independent direct action on pituitary gonadotrophs. This postulate is supported by *in vitro* studies using primary cultures of anterior pituitary cells from calves [30,45] and horses [31] wherein incubation with kisspeptin increased the level of LH in the culture medium. We planned our second experiment to address this line of thought–if the effect of peripheral kisspeptin treatment on LH release is independent of GnRH, then the blockage of GnRH receptors on gonadotrophs with an antagonist would not affect the kisspeptin-mediated LH release. However, we observed complete abolition of plasma LH peaks following kisspeptin treatment in heifers pre-treated with cetrorelix; thereby supporting the hypothesis that circulating kisspeptin causes LH release by a GnRH-mediated mechanism *in vivo*. Our results are similar to observations in the mouse and rat where LH secretion was suppressed by using GnRH receptor antagonists [4,15] and in sheep where disconnecting the pituitary from the hypothalamus prevented kisspeptin response [5,15,46]. A downstream effect of LH suppression in cetrorelix-treated heifers was also observed on the dominant follicle–all heifers failed to ovulate (0 ovulations of 5) in this group compared to 4 out of 5 heifers ovulating by 48 hours after kisspeptin treatment. Interestingly, the dominant follicle of all animals entered the regression phase by 72 hours after kisspeptin and GnRH antagonist treatment. A possible reason for the regression may be altered gene expression [47] and estradiol production by the dominant follicle [48] due to LH suppression [49].

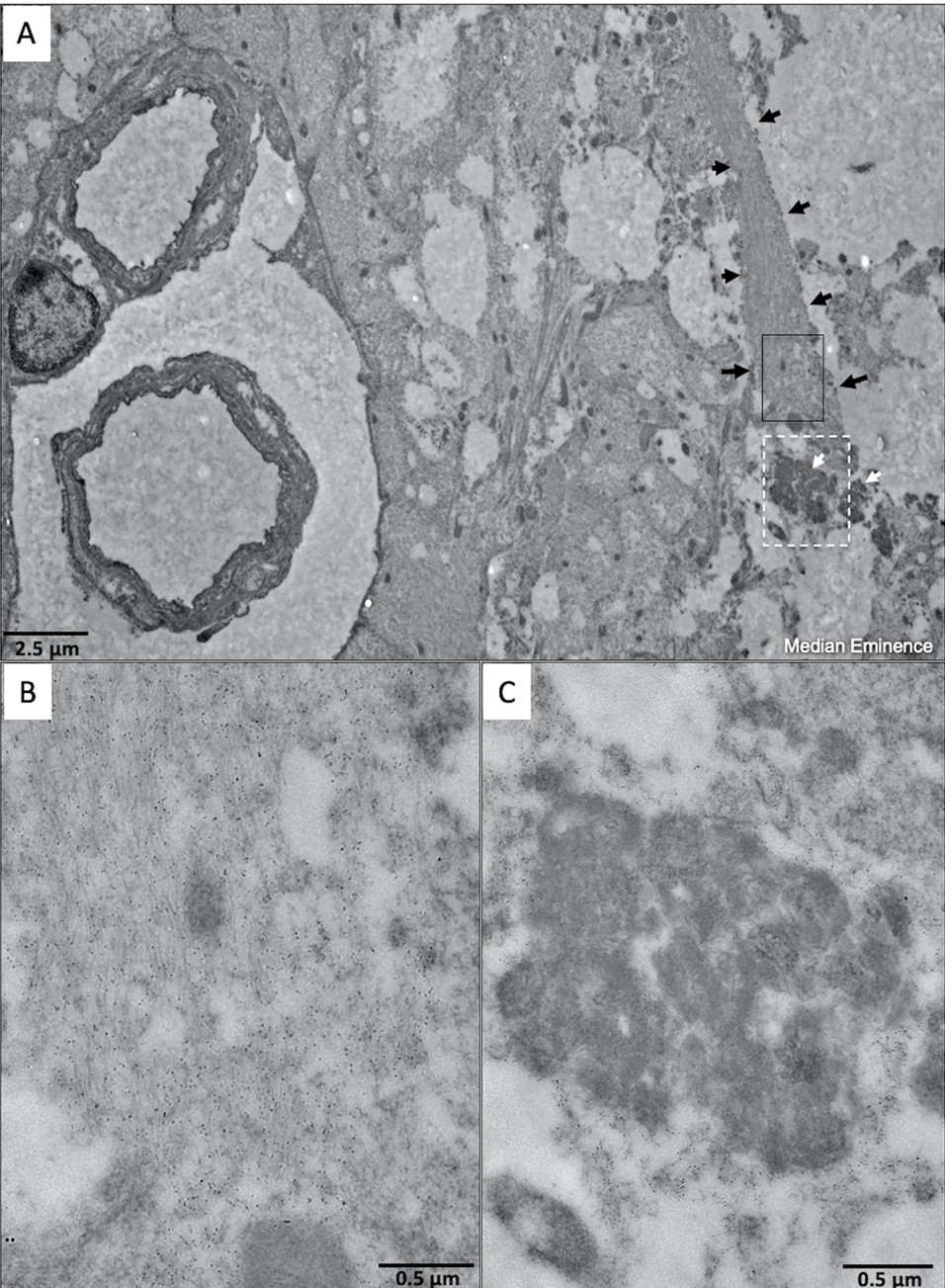

**Fig 4. Representative immuno-electron microscopy at the border of median eminence region of medio-basal hypothalamus in a proestrous cow after intravenous kisspeptin administration.** A) black arrows indicate synaptic terminals kisspeptin fiber projection, and white arrows indicate neurosecretory GnRH granules in axon terminals; B) Higher magnification from Fig A (black box) showing kisspeptin immunoreactive axon fiber projection is identified with gold particles (small black dots); C) Higher magnification from Fig A (dotted white rectangle) showing GnRH immunoreactive axon nerve terminal is identified with GnRH neurosecretory granules (DAB staining) surrounded by kisspeptin immunoreactivity (gold particles).

If circulating kisspeptin does not activate the GnRH neurons and causes GnRH-mediated LH release in cattle, the question still stands–how does peripheral injections of kisspeptin augment LH plasma concentrations? Kisspeptin neurons are clustered in two locations a cranial

**Table 3. Ovarian and endocrine responses (mean±SEM) of heifers treated with human Kisspeptin-10 (Kp10 group), pre-treatment with cetrorelix before Kisspeptin-10 (Cetrorelix group) or normal saline (Control group) intravenously under subluteal levels of plasma progesterone in Experiment 2.**

| Endpoint | Kp10 (n = 5) | Cetrorelix (n = 5) | Control (n = 5) | P-value* |
|---|---|---|---|---|
| Plasma progesterone on the day of treatment (Day 6) | | | | |
| | 1.86±0.43 | 1.41±0.56 | 1.86±0.38 | 0.73 |
| Ovulation rate after treatments: | | | | |
| number of heifer/total | 4/5[a] | 0/5[b] | 0/5[b] | 0.02 |
| hours after treatment[¶] | 48 | - | - | |
| Dominant follicle diameter (mm): | | | | |
| at time of treatment | 11.34±0.91 | 13.01±0.68 | 12.36±0.26 | 0.26 |
| 24h after treatment | 12.24±0.89 | 13.38±0.55 | 13.22±0.34 | 0.42 |
| 48h after treatment* | 16.51 | 12.91±0.55 | 14.67±0.56 | 0.06 |
| Number of ovulations after CIDR removal on Day 12: | | | | |
| Extant dominant follicle*# | 1/1 | 0/5 | 5/5 | 0.01 |
| New dominant follicle*† | 4/4 | 5/5 | 0/0 | 0.81 |

[a,b]Within rows, values with no common superscript are different (P≤0.05).

*Statistical analysis for two treatment groups (Cetrorelix versus Control).

[¶]Interval between treatment and ovulation.

#Dominant follicle present at the time of treatment.

†Dominant follicle of a new wave that emerged after treatment.

population in the medial preoptic area and second aggregation in the caudal portion of arcuate nucleus [50], and in cows the majority of kisspeptin neurons are detected in the rostral and middle regions of the arcuate nucleus [10,51]. In contrast, GnRH neurons are scattered in the diagonal band of Broca, the medial preoptic area and anterior hypothalamus [52], while the median eminence has a high density of GnRH positive nervefibers close proximity to kisspeptin fibers and portal blood vessels in cows (unpublished) and in primates [7]. The effect of peripheral kisspeptin on the GnRH cells could be exerted at the neuronal terminals in the median eminence (Fig 4) and/or organum vasculosum of the lamina terminalis, the target of GnRH secretion into the bloodstream. Additionally, the cFos protein expression is related to neuronal body stimulation and not necessarily axonal stimulation. As mentioned above, only the central administration of kisspeptin in mice induced cFOS expression in GnRH neurons which supported the idea that the stimulation at the perikarya is necessary to induce FOS. The terminals of the murine preoptic GnRH neurons axons reach the organum vasculosum of the lamina terminalis outside of blood-brain barrier [53]. That peripheral kisspeptin is acting at the median eminence is supported by the ability of kisspeptin to induce GnRH release *ex vivo* in sheep isolated media eminence, mouse mediobasal hypothalamus explants, and depolarization of GnRH neurons after exposed to kisspeptin onto GnRH nerve terminals [6,13,54]. This plausible explanation of release of pre-synthesized GnRH by action of circulating kisspeptin still remains to be tested by cause-and-effect experiments in cattle.

In conclusion, LH released in response to multiple intravenous doses of kisspeptin in cattle was mediated by a GnRH-dependent mechanism, but not through GnRH neuron activation as indicated by a failure to detect a treatment-induced increase in the proportions of cFOS positive GnRH immunoreactive neurons in the preoptic area and hypothalamus of cows. Pre-treatment with a GnRH antagonist, cetrorelix, 3 hours before initiation of intravenous treatments with kisspeptin suppressed the kisspeptin-induced LH surge, prevented ovulations, and caused regression of the extant dominant follicle followed by emergence of a new follicular wave, on average, 3.4 days after treatment in all five heifers. The likely mechanism of kisspeptin-

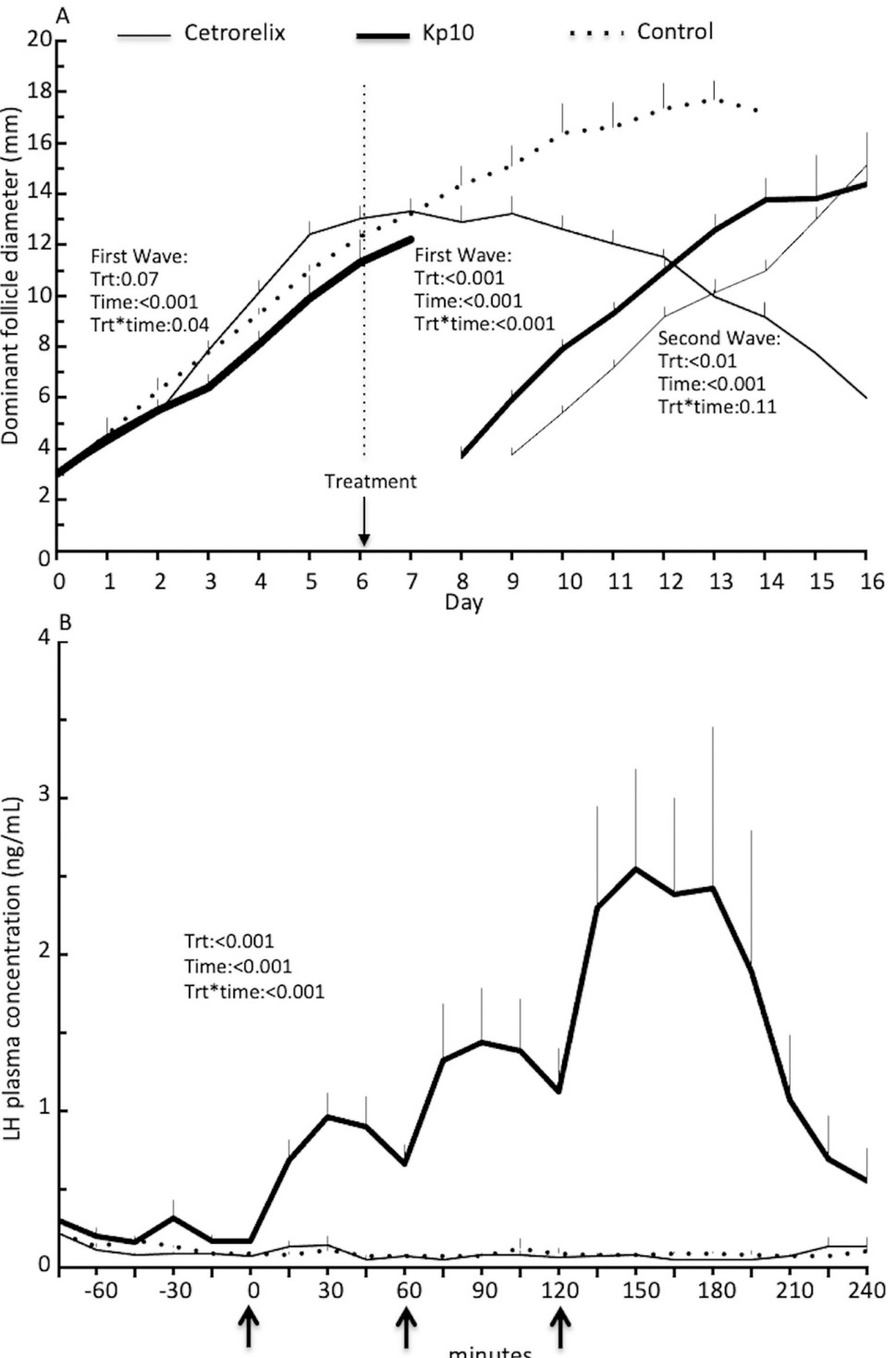

**Fig 5. Ovarian follicular dynamic and LH profile in heifers treated in the Experiment 2.** A) Growth profile of dominant follicle in pubertal heifers treated at Day 6 of follicular wave (day of wave emergence = Day 0) with human Kisspeptin-10 given as three injections of 15 mg at 60 minutes interval over a 2-hour period (Kp10 group, n = 5). Cetrorelix group (n = 5) was injected GnRH antagonist (Cetrorelix, 20 μg/Kg body weight, intramuscular) 3 hours before kisspeptin treatment while the Control group (n = 5) was given three intravenous injections of normal saline. B) Plasma LH concentrations in Kp10, Cetrorelix and Control group heifers.

mediated LH release is downstream of GnRH synthesis, perhaps by inducing release of pre-synthesized GnRH from the nerve terminals in the median eminence.

## Supporting information

**S1 Fig. Pre-absorption cFos antiboby to cFos protein. Fig A** low magnification of a cow hypothalamus section stained with Anti fos antibody. Arrow indicates the area shown in Fig B. **Fig B** high magnification of fos immunoreactive nuclei in cow hypothalamus (arrows). **Fig C** cow hypothalamus section stained with preabsorbed Fos antibody (1μg of antibody with 10 μg of fos protein). Arrow indicates the area shown in Fig D. **Fig D** High magnification of the area indicated by an arrow in Fig C showing no immunoreactivity.
(TIF)

## Acknowledgments

We thank Dr. O.J. Ginther (USA) for assistance with LH and progesterone radio-immunoassays, and L. Sobchishin from the WCVM Imaging Centre (Canada) for the help with electron microscope images. We also thank Isabel Franceschini from INRA (France) and Dr. R. Benoit from McGill University (Canada) for providing the anti-kisspeptin and anti-GnRH antibodies, respectively.

## Author Contributions

**Conceptualization:** Carlos E. P. Leonardi, Fernanda C. F. Dias, Gregg P. Adams, Jaswant Singh.

**Data curation:** Carlos E. P. Leonardi, Rodrigo A. Carrasco.

**Formal analysis:** Carlos E. P. Leonardi, Fernanda C. F. Dias, Jaswant Singh.

**Funding acquisition:** Jaswant Singh.

**Investigation:** Carlos E. P. Leonardi, Rodrigo A. Carrasco, Fernanda C. F. Dias, Eric M. Zwiefelhofer, Gregg P. Adams, Jaswant Singh.

**Methodology:** Carlos E. P. Leonardi, Rodrigo A. Carrasco, Fernanda C. F. Dias, Eric M. Zwiefelhofer, Gregg P. Adams, Jaswant Singh.

**Project administration:** Jaswant Singh.

**Supervision:** Gregg P. Adams.

**Validation:** Carlos E. P. Leonardi.

**Visualization:** Carlos E. P. Leonardi.

**Writing – original draft:** Carlos E. P. Leonardi, Rodrigo A. Carrasco, Fernanda C. F. Dias, Gregg P. Adams, Jaswant Singh.

**Writing – review & editing:** Carlos E. P. Leonardi, Rodrigo A. Carrasco, Fernanda C. F. Dias, Eric M. Zwiefelhofer, Gregg P. Adams, Jaswant Singh.

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
