## [Decision Letter · Decision Letter 0]

26 Sep 2022

PONE-D-22-22801Mechanism of LH release after peripheral administration of kisspeptin in cattlePLOS ONE

Dear Dr. Singh,

Thank you for submitting your manuscript to PLOS ONE. After careful consideration, we feel that it has merit but does not fully meet PLOS ONE’s publication criteria as it currently stands. Therefore, we invite you to submit a revised version of the manuscript that addresses the points raised during the review process.

We look forward to receiving your revised manuscript.

Kind regards,

Peter J. Hansen

Academic Editor

PLOS ONE

Journal Requirements:

Reviewers' comments:

Reviewer's Responses to Questions

**Comments to the Author**

1. Is the manuscript technically sound, and do the data support the conclusions?

Reviewer #1: Yes

Reviewer #2: Yes

2. Has the statistical analysis been performed appropriately and rigorously? 

Reviewer #1: Yes

Reviewer #2: Yes

3. Have the authors made all data underlying the findings in their manuscript fully available?

Reviewer #1: Yes

Reviewer #2: Yes

4. Is the manuscript presented in an intelligible fashion and written in standard English?

Reviewer #1: Yes

Reviewer #2: Yes

5. Review Comments to the Author

Reviewer #1: The manuscript by Leonardi et al. investigates the effects of exogenous administration of kisspeptin on LH secretion and ovulatory dynamics in cattle during the follicular phase of the estrous cycle. Additionally, authors investigate the effects of KP-10 on cFOS staining on GnRH neurons as a marker for neuronal activation. Finally, authors used electron microscopy to investigate close appositions between kisspeptin terminals and GnRH projections within the median eminence. In vivo studies were performed to characterize LH secretion, ovulatory response, and subsequent progesterone secretion in response to kisspeptin. To determine the role of GnRH on the kisspeptin-induced secretion of LH, authors included a group of heifers that was previously treated with a GnRH receptor antagonist. This is an interesting work using the bovine female as an animal model. Neuroendocrine research in cattle is scarce and findings in this species nicely complement the current knowledge body generated in primates, rodents, and sheep. The manuscript is well-written and easy to follow. The rationale for these studies is straightforward and well-described in the manuscript. Overall, the experiments appear to be well performed and described in good detail in the manuscript. Controls for immunohistochemistry procedures have been properly performed. Findings presented in the manuscript are novel and of importance relevance to the field of reproductive neuroendocrinology. The following points should be addressed during the revision:

Line 44 and throughout the manuscript: Since the ligand for the GPR54 has been identified, GPR54 has been renamed as Kiss1R. I suggest the authors could briefly mention here that GPR54 and Kiss1R are the same receptor and thereafter refer to the receptor as Kiss1R and not GPR54.

Line 48: “Furthermore, GnRH and kisspeptin immunoreactive cells are located in close association in the preoptic area and arcuate nucleus of the hypothalamus in mice, primates, sheep [7-9] and cattle [10].”. This statement is not technically correct. Are the authors referring to cell bodies or neuronal projections? In rodents, GnRH neurons are limited to the POA. In ruminants and primates, GnRH neurons migrate caudally to the anterior portions of the hypothalamus. In rare occasions, GnRH neurons can be found in the ARC in cattle, but the large majority of GnRH neurons is located in the POA and in the anterior hypothalamic area (AHA). If authors are referring to GnRH neuronal projections, those are also not commonly present in the ARC. In all species mentioned above, the large majority of GnRH projections are in the zona externa of the median eminence. Obviously, a large amount of kisspeptin projections is also observed in the median eminence, particularly in the internal zone.

Line 81: “…to stimulate GnRH neuronal cell bodies, acts on GnRH nerve terminals…” Remove “or”

Line 125: Please explain how the Kp10 dose (15 mg) was determined? Extrapolation from other species? Previous studies in cattle?

Line 219: While properly explained in the Methods, the design for Expt. 2 is somewhat hard to follow. A diagram figure with the experimental timeline would help readers to better understand the experimental design for study 2.

Line 223: PGF2alpha: Dose? Manufacturer? Mode of administration?

Line 322: Remove underline between “sections” and “from”

Line 323: what do authors mean with “association between kisspeptin and GnRH nerve terminals”? I suggest the authors to replace “association” with “close-appositions” or another technical term.

Line 392: remove “but” after “cows”

Line 412: “Currently, this approach is not technically feasible in cattle but a similar mechanism of action of kisspeptin is plausible.” That is not necessarily correct since third ventricle cannulation can be used in cattle to collect CSF and characterize secretion of GnRH and other hypothalamic neuropeptides. Please see papers by G Williams et al. (Biology of Reproduction, 1998; 59(3), 676-683 among other papers by the same group).

Line 434: replace “interesting” with “interestingly”

Line 442: “Kisspeptin neurons are clustered in two locations in cows – a cranial population in the medial preoptic area and second aggregation in the caudal portion of arcuate nucleus [50]” This is not technically correct. In cattle, majority of kisspeptin neurons are located in the rostral and middle regions of the ARC (see Tanco et al., 2016 PeerJ and Alves et al., BOR 2015).

Reviewer #2: The manuscript titled "Mechanism of LH release after peripheral administration of kisspeptin in cattle" investigates the mechanism whereby kisspeptin can act in cattle (lactating- Exp 1; pubertal heifers-Exp 2) to stimulate LH secretion, be it in the brain or at the pituitary. The authors are using dual immunohistochemistry in Exp 1 to pinpoint where iv kisspeptin acts to stimulate LH secretion (a dosing regimen of kisspeptin was used in a previous study which determined the dose needed to achieve adequate LH response). In addition the neuroanatomical analysis, the authors have also conducted an in vivo experiment to administer kisspeptin with and without a GnRH receptor antagonist, thereby determining if kisspeptin acts directly at the pituitary to stimulate LH secretion. The introduction and discussion are both logically sound. While there is considerable enthusiasm for this work, the concerns listed below should be addressed before full support of publication can be given by this reviewer.

Major comments:

1) The sample size in Experiment 1 is very low. Together with the variability of cFos between animals and greater than 50% more GnRH neurons identified in the treatment group, could the authors accurately conclude there is no increased activity of GnRH neurons with kisspeptin administration?

2) Do the authors think the use of lactating cows with low LH response (2 of 3) may be a confounding factor in the cFos experiment?

3) Please include references to validated use of kisspeptin and cFos antibodies in bovine hypothalamic tissue or provide details of preabsorption experiments herein. Lack of primary is not sufficient to determine antibody specificity. Also include details on these important controls in either the methods section or results section.

Minor comments:

1) italicize the use of in vivo, in vitro, ex vivo, etc throughout text of manuscript

2) Line 383- replace "neuron" with "cell bodies" given that you are arguing the neuron is activated at the terminal to release GnRH but not at the cell body

3) Make sure to include Cetrorelix + Kp10 in Table 3 and Figure 4

6. PLOS authors have the option to publish the peer review history of their article (what does this mean?). If published, this will include your full peer review and any attached files.

Reviewer #1: **Yes: **Rodolfo C. Cardoso

Reviewer #2: No

---

## [Author Response · Author response to Decision Letter 0]

10 Nov 2022

Reviewer #1:

1. Line 44 and throughout the manuscript: Since the ligand for the GPR54 has been identified, GPR54 has been renamed as Kiss1R. I suggest the authors could briefly mention here that GPR54 and Kiss1R are the same receptor and thereafter refer to the receptor as Kiss1R and not GPR54.

Thank you for the suggestion. Proposed change was done.

2. Line 48: “Furthermore, GnRH and kisspeptin immunoreactive cells are located in close association in the preoptic area and arcuate nucleus of the hypothalamus in mice, primates, sheep [7-9] and cattle [10].”. This statement is not technically correct. Are the authors referring to cell bodies or neuronal projections? In rodents, GnRH neurons are limited to the POA. In ruminants and primates, GnRH neurons migrate caudally to the anterior portions of the hypothalamus. In rare occasions, GnRH neurons can be found in the ARC in cattle, but the large majority of GnRH neurons is located in the POA and in the anterior hypothalamic area (AHA). If authors are referring to GnRH neuronal projections, those are also not commonly present in the ARC. In all species mentioned above, the large majority of GnRH projections are in the zona externa of the median eminence. Obviously, a large amount of kisspeptin projections is also observed in the median eminence, particularly in the internal zone.

Thank you for the detailed explanation. The sentence was change to “ Furthermore, GnRH and kisspeptin immunoreactive cells are located in the preoptic area and in the hypothalamus in mice, primates, sheep [7-9] and cattle [10], and its projection are detected in close association in the median eminence”.

3. Line 81: “…to stimulate GnRH neuronal cell bodies, acts on GnRH nerve terminals…” Remove “or”

It was removed

4. Line 125: Please explain how the Kp10 dose (15 mg) was determined? Extrapolation from other species? Previous studies in cattle?

The dose was determined based on our previous studies in cattle (Leonardi et al., 2018, Leonardi et al., 2020); “The dose and frequency of treatment was determined based on our previous studies in cattle [16, 17]”.

5. Line 219: While properly explained in the Methods, the design for Expt. 2 is somewhat hard to follow. A diagram figure with the experimental timeline would help readers to better understand the experimental design for study 2.

Thank you for the suggestion. Timeline diagram is now included as figure 1 in the revised manuscript

6. Line 223: PGF2alpha: Dose? Manufacturer? Mode of administration?

The sentence was add “(500 �g cloprostenol im, Estrumate, Merck Animal Health, Kirkland, QC, Canada)”

7. Line 322: Remove underline between “sections” and “from”

It was removed

8. Line 323: what do authors mean with “association between kisspeptin and GnRH nerve terminals”? I suggest the authors to replace “association” with “close-appositions” or another technical term.

It was replaced

9. Line 392: remove “but” after “cows”

It was removed

10. Line 412: “Currently, this approach is not technically feasible in cattle but a similar mechanism of action of kisspeptin is plausible.” That is not necessarily correct since third ventricle cannulation can be used in cattle to collect CSF and characterize secretion of GnRH and other hypothalamic neuropeptides. Please see papers by G Williams et al. (Biology of Reproduction, 1998; 59(3), 676-683 among other papers by the same group).

We were referring to hypophysial portal blood circulation. It is clarified in the revised manuscript.

11. Line 434: replace “interesting” with “interestingly”

It was fixed

12. Line 442: “Kisspeptin neurons are clustered in two locations in cows – a cranial population in the medial preoptic area and second aggregation in the caudal portion of arcuate nucleus [50]” This is not technically correct. In cattle, majority of kisspeptin neurons are located in the rostral and middle regions of the ARC (see Tanco et al., 2016 PeerJ and Alves et al., BOR 2015).

It was fixed

The sentence was added “and in cows the majority of kisspeptin neurons are detected in the rostral and middle regions of the arcuate nucleus [10, 51].” 

Reviewer #2:

1. The sample size in Experiment 1 is very low. Together with the variability of cFos between animals and greater than 50% more GnRH neurons identified in the treatment group, could the authors accurately conclude there is no increased activity of GnRH neurons with kisspeptin administration?

We understand the reviewer inquiry, however the number of GnRH cells that co-expressed cFOS is very low in both treatments. For this reason we believe that there is no effect of kisspeptin on the cFos activation.

2. Do the authors think the use of lactating cows with low LH response (2 of 3) may be a confounding factor in the cFos experiment?

Also there was not difference in the LH concentration among animals in the pre-treatment period. So, in our opinion, this is no confounding due to lactation in this experiment. We believe that this may be attributed to individual variation post-treatment which could happen in frequent-bleeding studies.

3. Please include references to validated use of kisspeptin and cFos antibodies in bovine hypothalamic tissue or provide details of preabsorption experiments herein. Lack of primary is not sufficient to determine antibody specificity. Also include details on these important controls in either the methods section or results section.

We performed the absorption validation as suggested. This is now included as supplementary data with analyses and images of pre-absorption cFos antiboby to cFos protein.

4. italicize the use of in vivo, in vitro, ex vivo, etc throughout text of manuscript

It was changed

5. Line 383- replace "neuron" with "cell bodies" given that you are arguing the neuron is activated at the terminal to release GnRH but not at the cell body

It was replaced

6. Make sure to include Cetrorelix + Kp10 in Table 3 and Figure 4 

Treatment of this group is described in the table and figure titles. So, we opted to keep the group name as cetrorelix. We hope that is OK with the reviewer.

---

## [Editor Report · Decision Letter 1]

15 Nov 2022

PONE-D-22-22801R1Mechanism of LH release after peripheral administration of kisspeptin in cattlePLOS ONE

Dear Dr. Singh,

Thank you for submitting your manuscript to PLOS ONE. After careful consideration, we feel that it has merit but does not fully meet PLOS ONE’s publication criteria as it currently stands. Therefore, we invite you to submit a revised version of the manuscript that addresses the points raised during the review process. Two changes are in order:I don't see any reference to the supplemental figure in the revised paper. Make sure you describe how the experiment was done in the methods and describe the results and cite Figure S1 in the Methods or Results section of  the paper.  My recommendation would be to include the figure with the paper itself and not as a supplemental Figure. If you do choose to make it supplementary, submit the figure as 1 pdf file that contains the figure and figure legend.  Send a copy of the manuscript that is not marked.  Please submit your revised manuscript by Dec 30 2022 11:59PM. If you will need more time than this to complete your revisions, please reply to this message or contact the journal office at plosone@plos.org. Please include the following items when submitting your revised manuscript:A rebuttal letter that responds to each point raised by the academic editor and reviewer(s). You should upload this letter as a separate file labeled 'Response to Reviewers'.A marked-up copy of your manuscript that highlights changes made to the original version. You should upload this as a separate file labeled 'Revised Manuscript with Track Changes'.An unmarked version of your revised paper without tracked changes. You should upload this as a separate file labeled 'Manuscript'.If applicable, we recommend that you deposit your laboratory protocols in protocols.io to enhance the reproducibility of your results. Protocols.io assigns your protocol its own identifier (DOI) so that it can be cited independently in the future. For instructions see: https://journals.plos.org/plosone/s/submission-guidelines#loc-laboratory-protocols. Additionally, PLOS ONE offers an option for publishing peer-reviewed Lab Protocol articles, which describe protocols hosted on protocols.io. Read more information on sharing protocols at https://plos.org/protocols?utm_medium=editorial-email&utm_source=authorletters&utm_campaign=protocols.

We look forward to receiving your revised manuscript.

Kind regards,

Peter J. Hansen

Academic Editor

PLOS ONE
---

## [Author Response · Author response to Decision Letter 1]

17 Nov 2022

We would like to thank the editor suggestions. The information about the protein used to pre-absortion with cFOS and immunostaining results are in the manuscript, however, we would like to keep the figure as supplementary data with the changes suggested by the editor (PDF file with figure and figure legend).

---

## [Editor Report · Decision Letter 2]

21 Nov 2022

Mechanism of LH release after peripheral administration of kisspeptin in cattle

PONE-D-22-22801R2

Dear Dr. Singh,

We’re pleased to inform you that your manuscript has been judged scientifically suitable for publication and will be formally accepted for publication once it meets all outstanding technical requirements.

Kind regards,

Peter J. Hansen

Academic Editor

PLOS ONE
---

## [Editor Report · Acceptance letter]

25 Nov 2022

PONE-D-22-22801R2 

Mechanism of LH release after peripheral administration of kisspeptin in cattle 

Dear Dr. Singh:

I'm pleased to inform you that your manuscript has been deemed suitable for publication in PLOS ONE. Congratulations! Your manuscript is now with our production department. 

Kind regards, 

on behalf of

Dr. Peter J. Hansen 

Academic Editor

PLOS ONE